# Reinforced education improves the quality of bowel preparation for colonoscopy: An updated meta-analysis of randomized controlled trials

**Xiaoyang Guo[1,2☯], Xin Li[3☯], Zhiyan Wang[3☯], Junli Zhai[3], Qiang Liu[1], Kang Ding[1], Yanglin Pan[iD][2]***

**1** Department of Ultrasound, The 305 Hospital of PLA, Bejing, China, **2** Xijing Hospital of Digestive Diseases, Air Force Medical University, Xi'an, China, **3** Department of Pneumology, The Second Medical Center of PLA General Hospital, Beijing, China

☯ These authors contributed equally to this work.

* yanglinpan@hotmail.com

**Data Availability Statement:** All relevant data are within the manuscript and its Supporting Information files.

## Abstract

### Background and aims

Inadequate bowel preparation (BP) is an unfavorable factor that influence the success of colonoscopy. Although standard education (SE) given to patients are proved useful to avoid inadequate BP. Studies concerning the effects of reinforced education (RE) on the quality of BP were inconsistent. The aim of this updated meta-analysis of randomized controlled trial was to compare the quality of BP between patients receiving RE in addition to SE and those receiving SE alone.

### Methods

MEDLINE, EMBASE, Web of Science and the Cochrane Library were systemically searched to identify the relevant studies published through April 2019. The primary outcome was the rate of adequate BP. Subgroup analyses were conducted. Secondary outcomes included BP score, adenoma detection rate (ADR), polyp detection rate (PDR), insertion time, withdrawal time, adverse events, >80% purgative intake and diet compliance. Dichotomous variables were reported as odds ratio (OR) with 95% confidence interval (CI). Continuous data were reported as mean difference (MD) with 95%CI. Pooled estimates of OR or MD were calculated using a random-effects model. Statistical heterogeneity was accessed by calculating the I2 value. A P value less than 0.05 was considered significant.

### Results

A total of 18 randomized controlled trails (N = 6536) were included in this meta-analysis. Patients who received RE had a better BP quality than those only receiving SE (OR 2.59, 95%CI: 2.09–3.19; P<0.001). A higher ADR (OR 1.35; 95%CI: 1.06–1.72; P = 0.020) and PDR (OR 1.24, 95%CI: 1.02–1.50; P = 0.030), shorter insertion (MD -0.76; 95%CI: -1.48-

**Funding:** All authors received no specific funding or salary for this work.

**Competing interests:** The authors have declared that no competing interests exist.

**Abbreviations:** ADR, adenoma detection rate; BBPS, Boston Bowel Preparation Scale; BMI, body mass index; BP, bowel preparation; CI, confidence interval; HCS, Harefield Cleansing Scale; MD, mean difference; OBPS, Ottawa Bowel Preparation Scale; OR, odd ratio; PDR, polyp detection rate; PEG, polyethylene glycol; RCT, randomized controlled trial; RE, reinforced education; SE, standard education; SMS, short message service; UBAS, Universal Bowel Assessment Scale.

(-0.04); P = 0.040) and withdrawal time (MD -0.83; 95%CI: -1.83-(-0.28); P = 0.003), less nausea/vomiting (OR 0.78; 95%CI: 0.64–0.97; P = 0.020) and abdominal distension (OR 0.72; 95%CI: 0.68–0.92; P = 0.020) were achieved in the RE group. More patients had >80% purgative intake (OR 2.17; 95%CI, 1.09–4.32; P = 0.030) and were compliant with diet restriction (OR 2.38; 95%CI: 1.79–3.17; P<0.001) in the RE group.

## Conclusion

RE significantly improved BP quality, increased ADR and PDR, decreased insertion and withdrawal time and adverse events.

## Introduction

Screening colonoscopies have been shown to decrease colorectal cancer incidence and mortality [1, 2]. High quality of bowel preparation (BP) is an essential factor of the success of colonoscopy. According to European Society of Gastrointestinal Endoscopy (ESGE) guideline, a ≥90% minimum standard for adequate BP was recommended [3]. However, about 18%-30.5% of the patients had an inadequate prepared colon in clinical practice [4, 5]. Inadequate BP leads to a higher rate of missed polyps or adenomas, increased healthcare cost, prolonged total procedural time and cancelled procedures [6–8]. According to recommendations from the US multi-society task force on colorectal cancer, patients should be provided with education instructions for all components of the colonoscopy preparation and emphasize the importance of compliance [9]. The latest ESGE guideline also recommended the use of enhanced instructions for BP [10].

In an effort to improve BP quality, researchers realized that regular oral or written instructions were insufficient and have focused on the strengthening of the instructions to patients before colonoscopy. In the early stage, additional explanations by a senior gastroenterologist were used to explain to patients according to their incorrect questionnaires [11]. Then, pictures [12], cartoon visual aids [13], booklets [14, 15] and even videos [16] were applied. With the development of economy and technology and the popularization of digital devices, phone call [17, 18], short message service [19, 20], smart phone applications [21, 22], social media [23] and online video [24] were employed to reinforce patients' education. However, the conclusions were inconsistent.

Four previous meta-analysis [25–28] have been published to systemically compare the adequacy of BP among patients receiving enhanced instructions and standard education. The design and search strategies of these studies were different. Chang et al's study [25] was the first meta-analysis determining the effect of educational intervention on BP quality. Although 9 RCTs (n = 2885) were included, three of them were abstracts. Desai et al [28] enrolled 6 studies, which only detected smartphone applications on BP quality compared with standard education (n = 810). Kurlander et al [26] enrolled 7 studies with full articles (n = 2660), however, two of them were not RCTs. In 2017, our team made comparisons of BP quality between patients receiving enhanced instructions plus regular instructions and regular instructions alone [27]. 8 RCTs (n = 3795) with full texts were enrolled. However, the literature search time were up to 2015. In the past 4 years, additional 10 high-quality clinical trials have been published [14, 16, 19–21, 24, 29–32]. The results, however, seemed to be conflicting. Therefore, here we further performed an updated meta-analysis to evaluate the influence of reinforced educations on the improvement of BP quality other outcomes.

## Methods

### Search strategies

We comprehensively searched Pubmed, EMBASE, Web of Science and the Cochrane Library through April 30, 2019. Only studies published in English were identified. Our key words and search strategies were as follows: 1, ("education" [All Fields] OR "educate" [All Fields]) AND ("colonoscopy" [All Fields] OR "colonoscopy" [MeSH]); 2, ("instruction" [All Fields] OR "instruct" [All Fields]) AND ("colonoscopy"[All Fields] OR "colonoscopy" [MeSH]); 3, ("education" [All Fields] OR "instruction" [All Fields]) AND ("bowel preparation" [All Fields] OR "bowel preparation" [MeSH] OR "bowel cleansing"); 4, ("instruction" [All Fields]) AND ("bowel preparation" [All Fields] OR "bowel preparation" [MeSH] OR "bowel cleansing" [All Fields]). In addition, reference lists of primary study publications, reviews, editorials and the proceedings of international congresses were manually searched. We did not consider abstracts or unpublished reports for inclusion.

### Study selection

The included studies were required to fulfill the following inclusion criteria: 1, study design: RCTs with full text; 2, study participants: patients ≥18 years old who underwent colonoscopy including both hospitalized patients and outpatients; 3, the primary or secondary outcomes included the rate of adequate BP; 4, study design: patients in the intervention group received reinforced educations by a certain of tool based on standard instruction, while patients in the control group received standard instructions; 5, there should be a qualified scale evaluating the degree of cleansing of colon. SE meant oral instructions, written instructions or oral plus written instructions associated with bowel preparation, which was provided by physicians or nurses before colonoscopy. The contents of SE included diet restriction, the time and methods of drinking purgatives. RE referred to additional, enhanced instructions based on SE, which was realized by providing some certain of methods or tools. The contents of SE and RE were generally the same.

### Study outcomes

The primary outcome was the rate of adequate bowel preparation. For the evaluation of BP quality, 5 BP scales were used, including Boston Bowel Preparation Scale (BBPS) [33], Ottawa Bowel Preparation Scale (OBPS) [34], Universal Preparation Assessment Scale (UPAS) [11], Harefield Cleansing Scale (HCS) [35] and Aronchick scale [36]. The adequacy of BP was defined by BBPS score ≥5, OBPS score <6, UBPS score <3 or HCS grade A or B. The secondary outcomes included BBPS or OBPS scores, adenoma or polyp detection rate (ADR or PDR), insertion time, withdrawal time and adverse events, >80% purgative intake and diet compliance.

### Data extraction

The studies were retrieved and the data were assessed and extracted by two investigators (Li X and Wang Z) independently, which was then summarized. Conflicts and disagreements were resolved by discussion or consulting a third investigator. Among each eligible study, the following data were extracted: author, year of publication, country, study design, blinding, number of patients allocated to each group, detailed information of interventions and controls, primary and secondary endpoints, BP scale, purgatives, diet restriction and other detailed information undergoing colonoscopy including insertion time, withdrawal time, ADR, PDR, purgative use, diet restrictions and so on.

## Quality assessment

Study quality was evaluated by modified Jadad's score [37, 38] (S2 Table), with 1–3 points being regarded as low quality and 4–7 points as high quality. Two studies used a nonrandom component in the sequence generation progress. Since patients were impossible to be blinded to instruction methods, all trials were single blinded to endoscopists, which may cause methodological impairment.

## Statistical analysis

All statistical analyses were performed using Review Manager (Revman, version 5.2) and Stata (version 12.0). If data from both intention-to-treat and per-protocol analyses were presented, the former were extracted and analyzed. Dichotomous data, including the rates of adequate bowel preparation, ADR or PDR, adverse events and diet compliance etc., were reported as odds ratio (OR) with 95% confidence interval (CI). Continuous data, including BBPS, OBPS, insertion and withdrawal time, were reported as mean difference (MD) with 95%CI. Pooled estimates of OR or MD were calculated using a random-effects model, in which both within-study and between-study variations were considered [39]. Subgroup analysis were conducted according to the types of RE (communicable or not), evaluation tool (BBPS or OPBS), indication (screening or mixed) and preparation method (4L PEG, split-dose or low-volume laxatives). Statistical heterogeneity was accessed by calculating the $I^2$ value, with substantial heterogeneity defined as $I^2$ greater than 50%, as described previously. A P value less than 0.05 was considered significant. Publication bias was assessed by visual inspection of a funnel plot using Review Manager and was detected by Stata software.

# Results

## Study selection

According to the predefined search strategies, a total of 1547 articles were identified initially. 787 records were removed due to duplications. Then, 730 articles were excluded after abstract reading. Of the remaining 30 articles, 12 were excluded after full-text reading for the following reasons: no BP quality as the primary or secondary outcomes (n = 2), non-RCTs (n = 5) and insufficient data (n = 5). Finally, 18 studies were included in this meta-analysis [11–24, 29–32] (Fig 1).

## Characteristics of the selected trials

The characteristics of 18 included studies were summarized in Table 1. A total of 6536 patients were enrolled. The pooled rete of adequate bowel cleansing was 81.0%, with 87.3% in the intervention group and 74.4% in the control group. Only two studies were multicenter studies [19, 23], the rest of which were conducted by single center. 17 trials' primary endpoint was BP quality, while one trial's primary endpoint was adherence with instruction [31]. Among all studies, secondary endpoints included: BP score, ADR or PDR, insertion time, withdrawal time etc.

There were some differences among these studies. Firstly, the quality of BP was evaluated by five scales. Secondly, methods that patients receiving REs were different (S1 Table). Thirdly, the type, volume and drinking methods of purgatives and diet restrictions were different.

For patients, one study [11] enrolled patients ≥40 years old and two studies [13, 21] ≥20, while the rest studies enrolled candidates with age ≥18 years old. Most trials took outpatients into consideration. One study only enrolled hospitalized patients [14] and patient type was unclear in other two [11, 17]. Furthermore, five studies [11–13, 16, 17] only enrolled patients

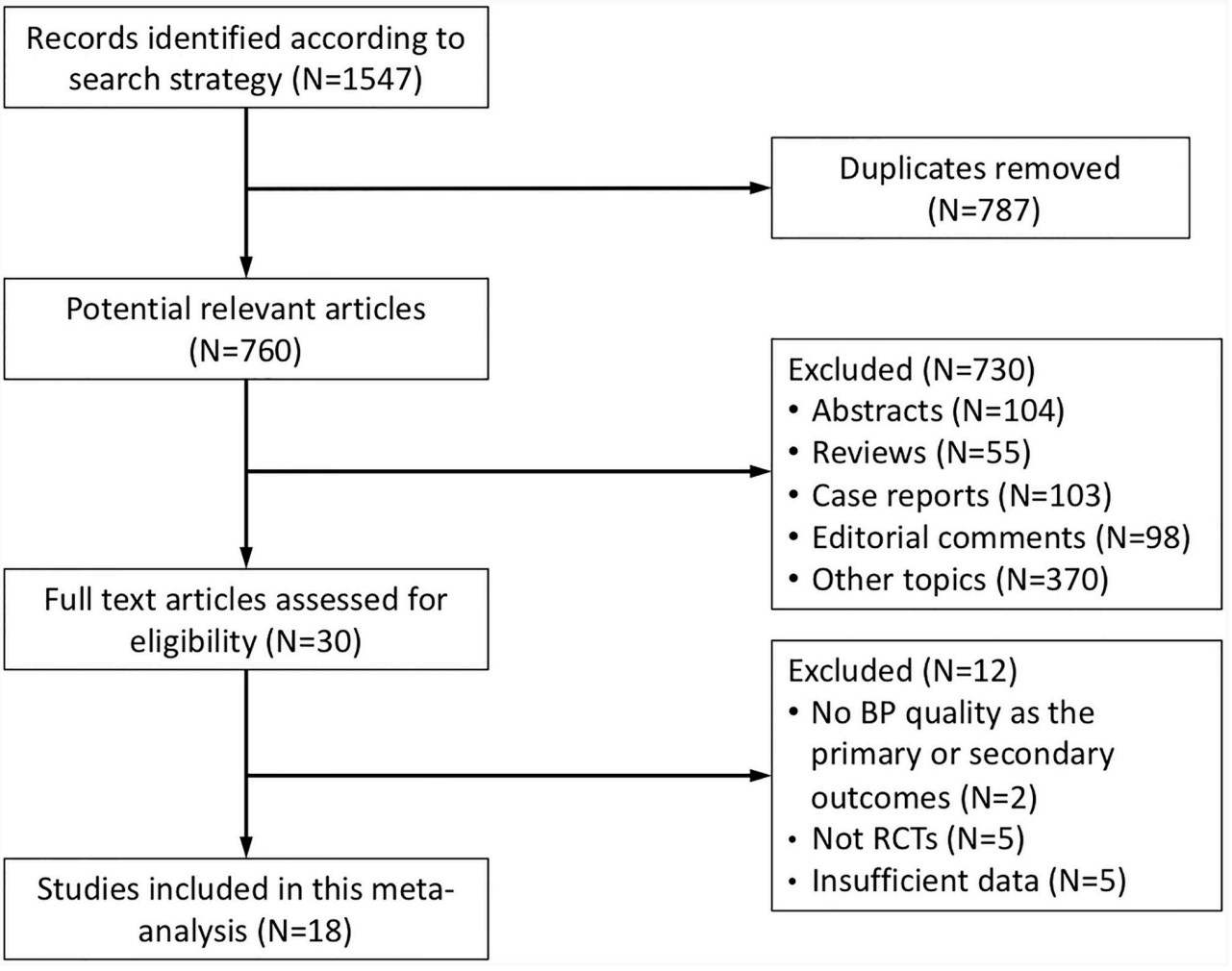

**Fig 1. Flow chart for search strategies.**

undergoing screening colonoscopy, while the others enrolled patients with mixed indications of colonoscopy, including screening, diagnosis and surveillance except one not reporting [31].

## Primary outcome: BP quality

As the primary outcome, the rates of BP quality were analyzed by all 18 studies (N = 6536) (Fig 2). In the RE group, 87.3% (2939/3366) of patients had adequate BP, while it was 74.4% (2359/3170) in the SE group (OR 2.59, 95%CI: 2.09–3.19; P<0.001).

## Subgroup analysis

**Communicable tool and non-communicable tool.** Generally, the means in the RE group can be divided into two kinds: communicable tool and non-communicable tool. 4 studies used communicable tools, including social media application (WeChat) [20, 23] and telephone call [17, 18]. Through the communicable tools, patients could communicate with physicians or nurses if they had any questions about bowel preparation during the procedure of instruction or met problems during the preparation. By using communicable tools for RE, patients

**Table 1. Characteristics of each included study.**

| | Design | Center | Blinding | Location | ITT Patient (RE/SE) | Patient | Primary endpoint | Indication | SE method | RE method | Communicable tools or not* |
|---|---|---|---|---|---|---|---|---|---|---|---|
| Back, 2018 [21] | RCT | Single | Single | Korea | 139/144 | Outpatient | BP quality | Mixed | Oral and leaflet | Audio-visual through smart phone | Yes |
| Calderwood, 2011 [12] | RCT | Single | Single | USA | 477/492 | Outpatient | BP quality | Screening | Written | Visual aid | No |
| Ergen, 2016 [14] | RCT | Single | Single | USA | 45/40 | Hospitalized patient | BP quality | Mixed | NR | Booklet | No |
| Elvas, 2016 [32] | RCT | Single | Single | Portugal | 116/113 | Outpatient | BP quality | Mixed | Oral and written | Additional personalized instruction | No |
| Kang, 2015 [23] | RCT | Multicenter | Single | China | 387/383 | Outpatient | BP quality | Mixed | Oral and written | Social media app | Yes |
| Lee, 2015 [17] | RCT | Single | Single | Korea | 253/137 | NR | BP quality | Screening | Oral and written | Telephone & SMS | Yes |
| Liu, 2014 [18] | RCT | Single | Single | China | 305/300 | Outpatient | BP quality | Mixed | Oral and written | Telephone | Yes |
| Liu, 2018 [29] | RCT | Single | Single | China | 239/237 | Outpatient | BP quality | Mixed | NR | Video plus retelling | No |
| Lorenzo, 2015 [22] | RCT | Single | Single | Spain | 108/152 | Outpatient | BP quality | Mixed | Written | Smart phone app | No |
| Modi, 2009 [11] | RCT | Single | Single | USA | 84/80 | NR | BP quality | Screening | Oral and written | Additional explanation | No |
| Park, 2015 [30] | RCT | Single | Single | Korea | 136/135 | Outpatient | BP quality | Mixed | Written | SMS | No |
| Park, 2016 [16] | RCT | Single | Single | Korea | 250/252 | Outpatient | BP quality | Screening | Written | Video | No |
| Rice, 2016 [24] | RCT | Single | Single | USA | 42/50 | Outpatient | BP quality | Mixed | Oral and written | Online video | No |
| Sharara, 2017 [31] | RCT | Single | Single | USA | 80/80 | Outpatient | Adherence with instructions | NR | Written | Smart phone app | No |
| Spiegel, 2011 [15] | RCT | Single | Single | USA | 216/220 | Outpatient | BP quality | Mixed | Oral, written | New designed booklet | No |
| Tae, 2012 [13] | RCT | Single | Single | Korea | 102/103 | Outpatient | BP quality | Screening | Verbal and written | Cartoon visual aids | No |
| Walter, 2019 [19] | RCT | Multicenter | Single | Germany | 248/247 | Outpatient | BP quality | Mixed | Oral | SMS | No |
| Wang, 2019 [20] | RCT | Single | Single | China | 257/127 | Outpatient | BP quality | Mixed | Written | WeChat & SMS | Yes |

ITT, intention to treat; RE, reinforced education; SE, standard education; BP, bowel preparation; RCT, randomized controlled trial; SMS, short message service; NR, not reported

* Communicable tools refer to the RE methods

achieved better BP quality (1035/1166, 88.8% vs. 678/914, 74.2%; OR 2.84; 95%CI:1.97–4.11; P<0.001). In other 14 studies with non-communication tools as the RE methods, patients also showed a higher rate of adequate BP compared with the control group (1902/2198, 86.5% vs. 1681/2254, 74.6%; OR 2.52; 95%CI:1.92–3.30; P<0.001) (S1 Fig).

**BBPS and OBPS.** 8 studies [12–14, 17, 19–21, 24] used BBPS to evaluate BP quality, a 10-point score from 0 to 9 (0 = very poor, 9 = excellent) by adding score of 3 segments of the

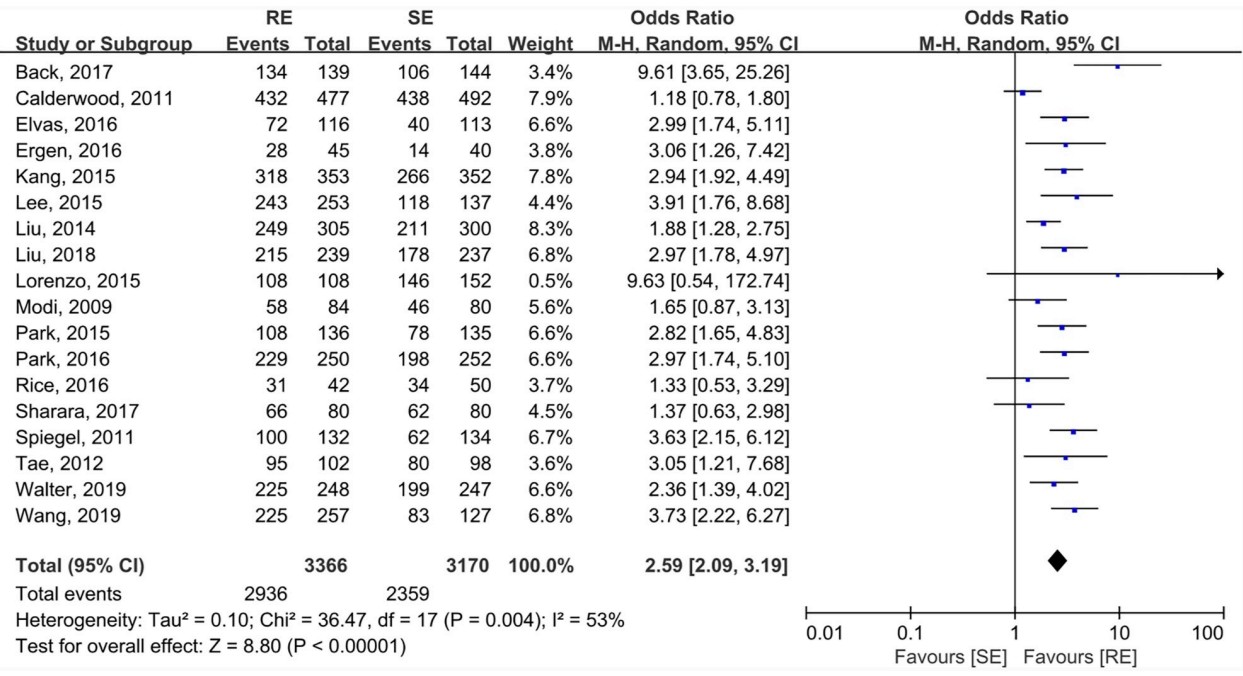

**Fig 2. Forest blot comparing the pooled BP quality between RE and SE groups.**

colon (right, transverse, and the left side of the colon), each of which was assigned a score ranging from 0 to 3 (0 = inadequate, 1 = fair, 2 = good, 3 = excellent). 3 studies [12, 13, 17] defined BBPS ≥5 as "adequate", two study [19, 20] set BBPS ≥6 as "adequate" and three studies [14, 21, 24] regarded a total BBPS ≥6 with all segment scores ≥2 as "adequate". Adequate rate of BP in RE group using BBPS was significantly higher than the controls (1413/1561, 90.5% vs. 1072/1333, 80.4%; OR 2.79; 95%CI:1.74–4.46; P<0.001). 6 studies [15, 16, 18, 19, 23, 30] used OBPS, which is calculated by adding the score of the right, transverse/descending, and sigmoid/rectum colon segments and fluid in the whole colon from 14 to 0 (14 = very poor, 0 = excellent). Adequate BP was defined as OBPS <6. Patients in RE group also showed a higher BP adequacy than those in SE group (1229/1424, 86.3% vs. 1014/1420, 71.4%; OR 2.61; 95%CI: 2.14–3.18; P<0.001) (S2 Fig).

**4L PEG in split-dose.** 6 studies [13, 14, 16, 23, 24, 30] used the purgative of 4L PEG in split-dose. It has been demonstrated that patients in RE group also showed a better BP quality than those in SE group (809/928, 87.2% vs. 670/927, 72.3%; OR 2.77; 95%CI: 2.16–3.55; P<0.001) (S3 Fig).

**Split-dose with any laxatives.** Among the included 18 studies, 12 used split-dose strategy. The laxatives included 3L [20] or 4L PEG [13, 14, 16, 21, 23, 24, 30], 2L PEG+Asc [17, 19, 21, 22] and SPMC [21, 31]. Patients with the administration of split dose in RE group showed better quality of bowel preparation compared with SE group (1810/2013, 90.0% vs. 1384/1814, 76.3%; OR 2.92; 95%CI: 2.31–3.68; P<0.001) (S4 Fig).

**Low-volume laxatives.** Several types of low-volume preparations have recently been shown with similar efficacy and lower adverse events compared with 4L PEG [10, 40]. Here 6 studies with 2492 patients used low-volume preparations, including 2L PEG+Asc [17, 19, 22], 2L PEG or NaP or magnesium citrate [15, 18, 29]. Compared with SE, RE showed higher rate of adequate BP in patients undergoing 2L PEG+Asc (576/609, 94.6% vs. 463/536, 86.4%; OR

2.84; 95%: 1.83–4.40; P<0.001) or 2L PEG (564/676, 83.4% vs. 451/671, 67.2%; OR 2.63; 95% CI: 1.75–3.97; P<0.001) (S5 Fig).

**Clear liquid diet and low fiber/residue diet.** In 7 studies [11, 14, 15, 18, 20, 23, 24], patients were only requested for dietary restriction of a clear liquid diet on the day before colonoscopy. Patients receiving RE indicated a better BP quality (1009/1218, 82.8% vs. 716/1083, 66.1%; OR 2.52; 95%CI: 1.90–3.35; P<0.001). In 6 studies [17, 19, 21, 22, 30, 32], patients were instructed to take low-fiber or low-residue diet 1–3 days before colonoscopy. Patient receiving RE also showed a higher BP quality (890/998, 89.2% vs. 687/926, 74.2%; OR 3.40; 95%CI, 2.35–4.92; P<0.001) (S6 Fig).

**Screening colonoscopy and mixed indications.** 5 studies [11–13, 16, 17] enrolled only patients undergoing screening colonoscopy. The BP quality of screening patients was better in the RE group than in the SE group (1057/1164, 90.8% vs. 880/1057, 83.3%; OR, 2.22; 95% CI, 1.35–3.67; P<0.001). 12 studies [14, 15, 18–24, 29, 30, 32] enrolled patients undergoing colonoscopy with mixed indications, including screening, diagnostic and surveillance. The BP quality of patients with mixed indications was better in the RE group than in the SE group (1813/2120, 85.5% vs. 1417/2031, 69.8%; OR 2.88; 95%CI: 2.33–3.55; P<0.001) (S7 Fig).

**SE methods.** The methods of SE were described in detail in 16 studies, including 6 with written instructions alone, 1 with oral instructions alone and 9 using written plus oral instructions. RE significantly improved the adequate rate in patients receiving only written instructions in the control group (1168/1308, 89.3% vs. 1005/1238, 81.2%; OR 2.29; 95%CI: 1.43–3.68; P<0.001). Similar results were found in patients receiving written plus oral instructions as the SE method (1300/1526, 85.2% vs. 963/1408, 68.4%; OR 2.77; 95%CI: 2.05–3.75; P<0.001) (S8 Fig).

## Secondary outcomes

**BBPS score and OBPS score.** 8 studies [12–14, 17, 19–21, 24] reported BBPS scores, and patients in RE group showed a higher BP score (mean score: 6.77 vs. 6.20; MD 0.72; 95%CI: 0.35–1.09; P<0.001). 5 studies [15, 16, 18, 23, 30] recorded OBPS scores, likewise, patients in the RE group had a lower BP score (mean score: 3.46 vs. 4.69; MD -0.66; 95%CI: -0.89-(-0.43); P<0.001) (S9 Fig).

**ADR and PDR.** ADR was reported in 4 studies [17, 20, 23, 30] and PDR was detected in 8 studies [12, 13, 16–18, 22, 29, 30]. Compared with those in the SE group, patients in the RE group had a higher ADR (226/1033, 21.9% vs. 135/782, 17.3%; OR 1.35; 95%CI: 1.06–1.72; P = 0.020) and PDR (637/2019, 33.2% vs. 483/1778, 28.2%; OR 1.24; 95%CI: 1.02–1.50; P = 0.030). Diminutive adenoma detection rate was reported in 1 study [23], which was also higher in the RE group (51/387, 13.2% vs. 30/383, 7.8%, P = 0.019) (S10 Fig).

**Insertion time and withdrawal time.** 8 studies [11–13, 16–18, 20, 23] reported insertion time and 9 studies [11–13, 16–18, 20, 23, 30] reported withdrawal time. Patient in RE group had a shorter insertion time (mean (min): 6.39 vs. 7.02; MD -0.76; 95%CI: -1.48-(-0.04); P = 0.040) and a shorter withdrawal time (mean (min): 7.23 vs. 8.02; MD -0.83; 95%CI: -1.83-(-0.28); P = 0.003) (S11 Fig).

**Adverse events.** 6 studies [12, 17, 18, 20, 23, 30] reported patients' adverse events after taking purgatives. The general rate of three main symptoms (nausea/vomiting, abdominal pain and abdominal distension) of adverse events was 12.1%. Patients receiving RE had less nausea/vomiting (339/1616, 21.0% vs. 301/1381, 21.8%; OR 0.78; 95%CI: 0.64–0.97; P = 0.020) and less abdominal distension (181/1751, 10.3% vs. 183/1516, 12.1%; OR 0.72; 95%CI: 0.68–0.92; P = 0.020). However, there was no statistical difference in abdominal pain between

patients in two groups (63/1616, 3.9% vs. 58/1381, 4.2%; OR, 0.99; 95%CI: 0.69–1.44; P = 0.970) (S12 Fig).

>**80% purgative intake and diet compliance.**   5 studies [17, 20, 21, 23, 30] reported the volume of purgatives that patients finally ingested. In RE group, more patients ingested >80% purgatives than those in SE group (1081/1172, 92.2% vs. 803/926, 86.7%; OR 2.17; 95%CI, 1.09–4.32; P = 0.030). 5 studies [17, 20, 23, 30, 31] reported diet compliance. Obviously, patients in the RE group were more compliant with diet restriction of the education (985/1079, 91.3% vs. 686/831, 82.6%; OR 2.38; 95%CI: 1.79–3.17; P<0.001) (S13 Fig).

## Sensitivity analysis

For the primary endpoint, the I2 value of heterogeneity was 53%. Sensitivity analysis was conducted with the extraction of study one by one. It showed that after extracting Calderwood's study and Back's study, the I2 changed to 30% and 44% separately, while after the extraction of other studies one by one, all I2 values were >50%.

## Publication bias

The funnel plots performed by Revman that was asymmetric (S14 Fig). Begg's test was conducted by Stata and the funnel plot showed no significant publication bias was found (P = 0.950) (S15 Fig).

## Discussion

Colonoscopy is an important preventive, diagnostic, and therapeutic modality, and its efficacy is closely associated with BP quality. Even though recommended by US Multi-society Task Force on Colorectal Cancer [9], standard oral or written instructions of BP before colonoscopy may still not be effective enough to ensure quality of BP, which leads to about 1/3 patients had inadequate BP [4, 41], far lower than the recommendation of a ≥90% minimum standard for adequate BP by ESGE guideline [3]. Therefore, investigators were hoping to improve BP quality through the enhancement of patients' education and multiple qualified RCTs have been conducted and reported. Apart from standard oral or written education, the reinforced education method is often more understandable, accessible or readable, which may improve patients' knowledge of BP, give patients a reminder before procedure and enable them to be more compliant with the instructions.

This updated meta-analysis including 18 qualified RCTs (N = 6536) with appropriate and variable reinforced educational methods, revealed that compared with SE, RE improves the quality of BP for colonoscopy (87.3% vs. 74.4%; P<0.001). For secondary outcomes, patients receiving RE had a better BP score, a higher ADR and PDR, shorter insertion time and withdrawal time, less nausea/vomiting and abdominal distension. Although the primary outcome was similar to the four previous systemic review and meta-analyses, this updated meta-analysis conducted some new conclusions in secondary outcomes: 1) RE improved both ADR and PDR, which firstly demonstrated that patients receiving RE had a higher PDR in the form of systemic review and meta-analysis; 2) patients in the RE group had a shorter insertion time; 3) less nausea/vomiting and abdominal distension were achieved in the RE group. In addition, this updated meta-analysis had the biggest sample size and the greatest number of qualified RCTs, which also included more kinds of reinforced education methods.

Among the included studies, RE methods or tools were variable. Desai et al [28] analyzed patients receiving RE by means of smartphone applications, which concluded that as a novel educational tool, smartphone application could achieve better bowel cleansing. However, in three studies, patients could not communicate with medical practitioners when met some

problems during BP period. In subgroup analysis of this meta-analysis, we divided RE tools into two kinds: communicable tools and non-communicable tools. Four studies used communicable tools, including a social media application (WeChat) [20, 23] and telephone call [17, 18]. Through the communicable tools, patients could communicate with physicians or nurses if they had any questions about bowel preparation during the procedure of instruction or met problems during the preparation. However, patients using non-communicable tools for communication are indirect and medical practitioners cannot receive feedbacks from patients until BP finished.

Detection and removal of adenomas and polyps is the most significant benefit of colonoscopy on the reduction of colorectal cancer mortality and morbidity [42]. This meta-analysis firstly demonstrated that RE could improve both ADR (OR 1.35; 95%CI: 1.06–1.72; P = 0.020) and PDR (OR 1.24; 95%CI: 1.02–1.50; P = 0.030). ADR was reported by four studies [17, 20, 23, 30]. Kang et al [23] showed that only diminutive adenomas (size ≤5mm) was significantly improved in RE group (13.2% vs. 7.8%, P = 0.019), while the size of adenomas was not described in other three trials. Although only a small group of diminutive adenomas (0.8%-3.8%) have advanced histological features [43], it is possible for diminutive adenomas to develop into advanced adenomas or cancers. PDR was reported in eight studies [12, 13, 16–18, 20, 29, 30] (33.2% vs. 28.2%; OR 1.24; 95%CI: 1.02–1.50; P = 0.030) and the conclusion was different from Guo et al's [27] and Chang et al's [25] study. Although having the biggest sample size among these studies, there was no difference between two groups in Calderwood et al's study (38.2% vs. 38.4%) [12]. However, Liu et al [18] showed that re-education through telephone had a higher PDR (38.0% vs. 24.7%).

This meta-analysis also showed that patients receiving RE had both shorter insertion time and withdrawal time which was different from our previous meta-analysis [27]. Although both BP examine (e.g adenoma, polyp and other colon disease) and BP evaluation were conducted when withdrawing, a colonoscopist may have a better visual when inserting colonoscopy, which could decrease insertion time. Generally, most discomforts for patients undergoing colonoscopy happened in insertion period, thus, to some extent, the decreasing of insertion time could relieve patient's pain and improve their willingness of colonoscopy.

In the past few years, several rating scales have been developed to evaluate the quality of BP, including BBPS [33], OBPS [34], UPAS [11], HCS [35], Aronchick scale [36] etc. BBPS is thought to be the best in clinical practice with high intra- and inter-observer reliability and good correlation with colonoscopic findings [9]. In this meta-analysis, BBPS was used in eight studies and OBPS in six studies. The subgroup analysis showed patients in EI group both had better BP quality no matter which evaluating methods (BBPS or OBPS) were used. For the secondary outcome of BBPS and OBPS scores, patients in RE also showed a better BBPS score.

Adverse events were reported in six studies [12, 17, 18, 20, 23, 30]. This meta-analysis firstly demonstrated that patients receiving RE had less nausea/vomiting or less abdominal distension, which is different from Guo et al's [27]. According to our conclusion, there was no difference between two groups with regard to abdominal pain and no heterogeneity ($I^2$ = 0%) of found in abdominal pain by sensitivity analysis. However, adverse events were influenced by some factors. Firstly, various purgatives were used, including 4L PEG in single [32] and split-dose [12, 14, 16, 23, 24, 30], 3L PEG [20], 2L PEG [15, 17–19, 22, 29]. Secondly, nausea and vomiting were combined for analysis in three studies [12, 18, 23] and analyzed separately in two studies [17, 20]. Thirdly, some detailed information was not provided in these studies, such as drinking speed, time interval between starting taking purgative and adverse events happening. Further studies may control and eliminate these interference factors.

Although our finding confirms the effectiveness of RE in BP, there are some limitations and several areas worth further investigation. Firstly, only two studies were conducted by

multicenter, the rest of which were carried out in single center. Secondly, owing to the superiorities and weaknesses of different tools of educations, the head-to-head comparison of different RE methods needs to be further investigated. It is possible that a combination use of two or more means of RE methods could achieve better BP quality. Thirdly, for the patients with high risk factors associated with inadequate BP (e.g the elderly, BMI $\geq$25 or constipation), adequate BP is less likely to be achieved. RE for such patients may be even more significant for a better BP quality. Further work needs to be done to investigate the impacts of RE in these patients. Fourthly, patients with younger age, without comorbidities (especially constipation, diabetes, Parkinson disease and spine injury) or medications (especially tricyclic antidepressant (TCA) and possibly calcium channel blockers (CCB) and those with higher education level represent an relatively "easy-to-prepare" group [44]. It is interesting to investigate whether these patients may achieve adequate bowel preparation even without RE. Unfortunately, among the 18 studies focusing on investigating the effects of RE on BP quality, none reported the results of bowel preparation in low-risk patients. Further studies are needed to investigate the effects RE on BP quality in the "easy to-prepare" population. Fifthly, among the 5 scales used for the evaluation of BP quality, only BBPS [33, 45] and OBPS [23, 34] were validated for the inter- and intra-observer consistence. Uncertainty of the results may exist with the uses of other 3 invalidated scales. Last but not the least, to evaluate the effects of RE in different conditions, several subgroup analyses were performed. Although significant differences were found in most of the analyses (P<0.001). The power of the subgroup analyses may be not sufficient. The capabilities of subgroup analyses to detect meaningful differences between studies is often limited, thus it should be cautioned to explain the results.

In summary, this updated meta-analysis indicated that compared with SE, RE could significantly improve BP quality, increase ADR and PDR, decrease insertion time and withdrawal time and reduce adverse events of nausea/vomiting and abdominal distension. Therefore, in addition to SE, RE before colonoscopy should be recommended for patients undergoing BP.

## Supporting information

**S1 Fig. Subgroup: Communicable and non-communicable tool.**
(TIFF)

**S2 Fig. Subgroup: BBPS and OBPS.**
(TIFF)

**S3 Fig. Subgroup: 4L PEG in split dose.**
(TIFF)

**S4 Fig. Subgroup: Split dose with any laxative.**
(TIFF)

**S5 Fig. Subgroup: Low volume laxative.**
(TIFF)

**S6 Fig. Subgroup: Clear liquid diet and low fiber/residue diet.**
(TIFF)

**S7 Fig. Subgroup: Screening colonoscopy and mixed indications.**
(TIFF)

**S8 Fig. Subgroup: Written and oral plus written instructions.**
(TIFF)

**S9 Fig. Secondary outcome: A, BBPS score; B, OBPS score.**
(TIFF)

**S10 Fig. Secondary outcome: A, ADR; B, PDR.**
(TIFF)

**S11 Fig. Secondary outcome: A, insertion time; B, withdrawal time.**
(TIFF)

**S12 Fig. Secondary outcome: Adverse events.**
(TIFF)

**S13 Fig. Secondary outcome: A, >80% purgative intake; B, diet compliance.**
(TIFF)

**S14 Fig. Funnel blot indicating no significant publication bias (Revman).**
(TIFF)

**S15 Fig. Funnel blot indicating no significant publication bias (Stata).**
(TIFF)

**S16 Fig. Forest plot comparing BP quality between RE and SE group (Stata).**
(TIFF)

**S1 Table. Details of each included study.**
(DOCX)

**S2 Table. Details of quality assessment using modified Jadad score.**
(DOC)

**S3 Table. Bowel preparation quality in patients receiving split-dose with any laxatives.**
(DOCX)

**S1 Checklist. PRISMA 2009 checklist to be included with meta-analyses.**
(DOC)

## Author Contributions

**Conceptualization:** Yanglin Pan.

**Data curation:** Xin Li, Zhiyan Wang, Junli Zhai.

**Formal analysis:** Qiang Liu, Kang Ding.

**Methodology:** Xiaoyang Guo.

**Software:** Xiaoyang Guo.

**Writing – original draft:** Xiaoyang Guo.

**Writing – review & editing:** Yanglin Pan.

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
