## [Decision Letter · Decision Letter 0]

5 Dec 2019

PONE-D-19-27355

Reinforced education improves the quality of bowel preparation for colonoscopy: an updated meta-analysis of randomized controlled trials

PLOS ONE

Dear Dr. Yanglin Pan,

Thank you for submitting your manuscript to PLOS ONE. After careful consideration, we feel that it has merit but does not fully meet PLOS ONE’s publication criteria as it currently stands. Therefore, we invite you to submit a revised version of the manuscript that addresses the points raised during the review process.

Comments from the Editor

A rigorous statistical analysis is clue in this type of manuscripts. I agree with the statistician that it needs improvement and better describe in more detail. Please pay a very careful attention to the statistician comments. As mentioned by the statistician, currently the manuscript do not meet the Plos One criteria (see publication criteria number 3).I consider also that the comments from the rest of the reviewers are important in order to improve the manuscript.Flow Chart. From 760 manuscripts, 523 were excluded, and then 27 were assessed for eligibility, however, 760-523 are not 27. Please correct or explain it.Is the Universal preparation scale a validated scale? This is a limitation if not.Results (primary and secondary outcomes). Please include the confidence intervals of the proportions (not only of the OR)Please English language needs some improvement. Make sure that the manuscript is edited.In page 11, the paragraph “and the conclusion ….until (38.2% vs. 38.4%)" needs to be rephrased

We would appreciate receiving your revised manuscript by January 15th 2020. To enhance the reproducibility of your results, we recommend that if applicable you deposit your laboratory protocols in protocols.io, where a protocol can be assigned its own identifier (DOI) such that it can be cited independently in the future. For instructions see: http://journals.plos.org/plosone/s/submission-guidelines#loc-laboratory-protocols

We look forward to receiving your revised manuscript.

Kind regards,

Antonio Z Gimeno-Garcia

Academic Editor

PLOS ONE

Journal Requirements:

2. We noticed you have some minor occurrence(s) of overlapping text with the following previous publication(s), which needs to be addressed:

https://doi.org/10.1016/j.gie.2016.05.012

In your revision ensure you cite all your sources (including your own works), and quote or rephrase any duplicated text outside the Methods section. Further consideration is dependent on these concerns being addressed.

'No financial Disclosure'

Please provide an amended Funding Statement that declares *all* the funding or sources of support received during this specific study (whether external or internal to your organization) as detailed online in our guide for authors at http://journals.plos.org/plosone/s/submit-now.  

Please state what role the funders took in the study.  If any authors received a salary from any of your funders, please state which authors and which funder. If the funders had no role, please state: "The funders had no role in study design, data collection and analysis, decision to publish, or preparation of the manuscript."

Additional Editor Comments (if provided):

Reviewers' comments:

Reviewer's Responses to Questions

**Comments to the Author**

1. Is the manuscript technically sound, and do the data support the conclusions?

Reviewer #1: Yes

Reviewer #2: Yes

Reviewer #3: Partly

2. Has the statistical analysis been performed appropriately and rigorously? 

Reviewer #1: Yes

Reviewer #2: Yes

Reviewer #3: No

3. Have the authors made all data underlying the findings in their manuscript fully available?

Reviewer #1: Yes

Reviewer #2: Yes

Reviewer #3: Yes

4. Is the manuscript presented in an intelligible fashion and written in standard English?

Reviewer #1: Yes

Reviewer #2: No

Reviewer #3: Yes

5. Review Comments to the Author

Reviewer #1: I congratulate the authors for this updated review and meta-analysis that I have enjoyed reading. The paper is well written, the results and conclusions are sound.

Nevertheless, I have some questions, comments and suggestions.

Major

1. Split dose.

I miss an analysis of studies grouped by the split-dose with any laxative. Split-dosing is the variable that influences most the quality of bowel cleansing and it has been showed that it is not dependent of the laxative used.

Is it possible to group the studies regarding just the split-dose with any laxative?

2. I miss an exploration of the population included. There are some studies that include a population with no risk factors of poor bowel preparation (e.g. young, with no comorbidities or medications that may impair the bowel peristalsis) or a population “easy to educate” (e.g. young, high education, technologically competent).

Is it possible to group the studies regarding the population included? It will allow to compare the studies and to look if the conclusions of those studies may be generalized to the general population.

If it is not possible to analyze the population included I will like to include that as a limitation.

3. The quality of the standard education varies in the different studies. It is possible that if the quality of the SE is low, such as written instructions only, the room for improvement may be greater than if the quality of the SE is better. Is it possible to group the studies in relation with the quality of the SE?

If it is not possible I will appreciate a comment in the limitations section.

Minor

1. Originality. As the authors have stated there have been 4 previous meta-analysis exploring the association between an enhanced instruction for bowel prep and the bowel cleansing adequacy. Nevertheless the present study is the most updated analysis and it includes several RCTs in the last 4 years.

2. There is a discrepancy between the 2 abstracts, in the first, line 8, it is stated that the literature search was done through March 2019. In the manuscript abstract, page 3, line 9, it says that the search was conducted until April 2019. Please correct it.

4. Page 6, lines 1,2 “diet restriction” is duplicated.

5. The effect of a reinforced education may be greater when the interval between the SE and the bowel preparation or the colonoscopy is longer, especially when the RE is made with a communicable tool. Is it possible to group the studies or the individual data to asses that variable?

Reviewer #2: MAJOR

1) The authors need to improve the language in the manuscript. There are spelling mistakes (pay attention to the capital letters in the names, et al needs a dot after the word, like that “et al.”), and some difficult to understand sentences. For example:

i. Page 6, line 2 withdraw. You should change it for withdrawal.

ii. Page 6, line 6. You should change it for rate. I also suggest to add in this sentence, “The pooled rate of adequate bowel cleansing was …”

2) The figures and tables legend is not complete. Moreover, the references of the figures and supplementary figures don´t match in the text.

3) The authors have missed some articles in this meta-analysis, which fulfill the inclusion criteria. Do you have any reason to exclude them?

a. Galvez M, Zarate AM, Espino H, Higuera-de la Tijera F, Awad RA, Camacho S. A short telephone-call reminder improves bowel preparation, quality indicators and patient satisfaction with first colonoscopy. Endosc Int Open. 2017;5. E1172-E8.

b. Elvas L, Brito D, Areia M, Carvalho R, Alves S, Saraiva S, et al. Impact of Personalised Patient Education on Bowel Preparation for Colonoscopy: Prospective Randomised Controlled Trial. GE Port J Gastroenterol. 2017;24:22---30.

c. Prakash SR, Verma S, McGowan J, Smith BE, Shroff A, Gibson GH, et al. Improving the quality of colonoscopy bowel preparation using an educational video. Can J Gastroenterol. 2013;27:696---700.

MINOR

4) It could help to the reader to explain the definition of standard instructions in the study selection.

5) Pag 7, paragraph 1. It could be better to add the reference after each bowel preparation scale to let the reader know which are the studies are you referring to.

6) Pag 7, communicable and non-communicable tools. I suggest to add the definition and the reference to the correct supplementary table.

7) Pag 8- Did you analyze the BP quality by using other purgatives as low volume preparations?

8) It could be better to add in the results section some data regarding the diminutive adenoma detection rate as you discussed it latter.

Reviewer #3: Thank you for the opportunity to review this paper. This is an interesting manuscript presenting a results of a meta-analysis of randomized controlled trials on reinforced education improves the quality of bowel preparation for colonoscopy.

My review mainly concerns only the statistical aspects of the study. Some questions reported below were raised and in my view, it is not acceptable in this version for the publication in this journal.

Comments

1. Statistical analysis paragraph is a duplicate of the paragraph of the precedent paper [Guo X, Yang Z, Zhao L, et al. Enhanced instructions improve the quality of bowel preparation for colonoscopy: a meta-analysis of randomized controlled trials. Gastrointest Endosc 2017;85:90-97 e6.]. More detail are necessary. The meta-analysis was conduct on OR and on mean differences in the statistical analysis paragraph should be detailed.

2. Why in Figure 2 the OR estimated for Lee,2015 is different to the estimate of the previous paper? (3.91 vs 4.38)

3. There are many subgroup analysis the capabilites of subgroup analyses to detect meaningful differences between studies is often limited. Subgroup analyses also need sufficient power.

Minor comments

• There are many error of typing. Please, verify the manuscript for English and typos;

• p-value should be reported with three decimal number and with exact value; uniform the notation or P or p

• In the funnel plot the line of the 95% CI can make easier to read the figure

6. PLOS authors have the option to publish the peer review history of their article (what does this mean?). If published, this will include your full peer review and any attached files.

Reviewer #1: Yes: Marco A Alvarez-Gonzalez MD PhD

Reviewer #2: No

Reviewer #3: No

---

## [Author Response · Author response to Decision Letter 0]

23 Jan 2020

Dear editor and reviewers,

We thank the editor and reviewers for reviewing our Manuscript ID PONE-D-19-27355 entitled “Reinforced education improves the quality of bowel preparation for colonoscopy: an updated meta-analysis of randomized controlled trials”. We have revised the manuscript according to the constructive suggestions. A point-by-point response is included below. 

In response to the reviewers' comments, additional information has been added to the revised manuscript. These are marked by red highlights in the version with tracked changes. The version with all changes accepted (clean version) are also provided. Deleted text no longer appears in the final clean version.

Editor

1, A rigorous statistical analysis is clue in this type of manuscripts. I agree with the statistician that it needs improvement and better describe in more detail. Please pay a very careful attention to the statistician comments. As mentioned by the statistician, currently the manuscript do not meet the Plos One criteria (see publication criteria number 3).

Response: Thank you for the advice. We totally agree with reviewer 3 that statistical methods should be described in more detail. 

According to the suggestion, the methods of Statistical analysis were modified as follows (page 6, line24–page 7, line 6): “All statistical analyses were performed using Review Manager (Revman, version 5.2) and Stata (version 12.0). If data from both intention-to-treat and per-protocol analyses were presented, the former were extracted and analyzed. Dichotomous data, including the rates of adequate bowel preparation, ADR or PDR, adverse events and diet compliance etc., were reported as odds ratio (OR) with 95% confidence interval (CI). Continuous data, including BBPS, OBPS, insertion and withdrawal time, were reported as standard mean difference (SMD) with 95%CI. Pooled estimates of OR or SMD were calculated using a random-effects model (M-H, heterogeneity), in which both within-study and between-study variations were considered (Control Clin Trials 1986;7:177-88.). Subgroup analysis were conducted according to the types of RE (communicable or not), evaluation tool (BBPS or OPBS), indication (screening or mixed) and preparation method (4L PEG, split-dose or low-volume laxatives). Statistical heterogeneity was accessed using the Cochrane Q test and by calculating the I2 value, with substantial heterogeneity defined as I2 greater than 50%, as described previously. A P value less than 0.05 was considered significant.”

 We believe the current version of statistical methods should meet the publication criteria No.3 of PLoS One.

2, I consider also that the comments from the rest of the reviewers are important in order to improve the manuscript.

Response: Thank you for the suggestion. The constructive and suggestive comments from the reviewers and the editor are very helpful for improving our manuscript. The manuscript had been revised based on the suggestions. Point-to-point responses had been provided for all the questions and comments.

3, Flow Chart. From 760 manuscripts, 523 were excluded, and then 27 were assessed for eligibility, however, 760-523 are not 27. Please correct or explain it.

Response: Thank you for pointing out the mistake. We performed systemic literature search for several times. The outcomes of each time of searching may be mixed. We have corrected the flow chart.

4, Is the Universal preparation scale a validated scale? This is a limitation if not.

Response: Thank you for the comments. We agree that the use of invalidated UPAS scale to evaluate BP quality may bring some uncertainty to the results. The following contents were added into Limitation (page 14, line 17-19): “Fifthly, among the 5 scales used for the evaluation of BP quality, only BBPS (Gastrointest Endosc 2009;69:620-5; Gastrointest Endosc 2010;72:686-92;) and OBPS (Clin Gastroenterol Hepatol 2016;14:429-435 e3; Gastrointest Endosc 2004;59:482-6) were validated for the inter- and intra- observer consistence. Uncertainty of the results may exist with the uses of other 3 invalidated scales.” 

5, Results (primary and secondary outcomes). Please include the confidence intervals of the proportions (not only of the OR)

Response: Thank you for the suggestion. We have added confidence intervals to all results.

6, Please English language needs some improvement. Make sure that the manuscript is edited.

Response: Thank you for the suggestion. The manuscript had been thoroughly reviewed and modified by a native English editor. The grammatical errors and typing errors had been carefully checked and corrected.

7, In page 11, the paragraph “and the conclusion ….until (38.2% vs. 38.4%)" needs to be rephrased

Response: Thank you for the question. There was no statistical difference in polyp detection rate (PDR) between patients receiving enhanced instructions and regular instructions (448/1245, 36.0% vs 363/1179, 30.8%; OR, 1.25; 95% CI, 0.93-1.68; P=0.140) either in Guo et al’s (Gastrointest Endosc 2017;85:90-97) meta-analysis or in Chang et al’s meta-analysis (Endosc Int Open 2015;3:E646-52) (RR, 1.14; 95% CI, 0.87-1.51). Likewise, Calderwood et al’ s study also demonstrated no statistical difference between intervention and control groups (182/477, 38.2% vs. 189/492, 38.4%; P=0.930). However, this updated meta-analysis showed opposite conclusion that patients receiving reinforced education had a higher PDR (637/2019, 33.2% vs. 483/1778, 28.2%; OR 1.24; 95%CI: 1.02-1.50; P=0.030) compared with those receiving regular educations. It may because this meta-analysis had enrolled more studies and patients.

Reviewer #1: 

Major

1. Split dose. I miss an analysis of studies grouped by the split-dose with any laxative. Split-dosing is the variable that influences most the quality of bowel cleansing and it has been showed that it is not dependent of the laxative used. Is it possible to group the studies regarding just the split-dose with any laxative?

Response: Thank you for the question. We agree with the opinion of the reviewer that split-dose is an important factor associated with the quality of bowel preparation. It deserves further investigation with split-dose with any laxatives as one subgroup. We added the contents into Results as follows (page 9, line 6-10): “Split-dose with any laxatives. Among the included 18 studies, 12 used split-dose strategy. The laxatives included 3L or 4L PEG (n=8), 2L PEG+Asc (n=4) and SPMC (n=2). Patients with administration of split-dose laxatives in RE group showed better quality of bowel preparation compared with SE group (1810/2013, 90.0% vs. 1384/1814, 76.3%; OR 2.92; 95%CI: 2.31-3.68; P<0.001).”

The results of the subgroup analysis were supplemented as supplementary table 3, which was provided as follows.

Supplementary table 3. Bowel preparation quality in patients receiving split-dose with any laxatives.

 Studies Adequate rate of bowel preparation BBPS score

 RE SE 95%CI P value Mean score SMD 95%CI P value

Split-dose with any laxatives 12 90.0% 76.3% 2.31-3.68 <0.001 6.71 vs. 6.23 1.81 0.50-3.12 0.007

3L or 4L PEG 8 88.2% 71.7% 2.40-4.06 <0.001 6.95 vs. 6.07 0.69 0.13-2.09 0.020

2L PEG+Asc 4 94.9% 83.7% 2.08-8.77 <0.001 7.23 vs. 6.39 3.38 0.15-6.61 0.040

SPMC 2 91.3% 75.0% 0.51-24.66 0.200 / / / /

Abbreviations: PEG, polyethylene glycol; RE, reinforced education; SE, standard education; SMD, standard mean difference; CI, confidence interval; Asc, ascorbic acid; SPMC, sodium picosulfate with magnesium citrate

2. I miss an exploration of the population included. There are some studies that include a population with no risk factors of poor bowel preparation (e.g. young, with no comorbidities or medications that may impair the bowel peristalsis) or a population “easy to educate” (e.g. young, high education, technologically competent). Is it possible to group the studies regarding the population included? It will allow to compare the studies and to look if the conclusions of those studies may be generalized to the general population. If it is not possible to analyze the population included I will like to include that as a limitation.

Response: Thank you for the suggestion. There are several patients-related parameters which had been identified as risk factors associated with higher quality of bowel preparation in the past decade. As the reviewer indicated, patients with younger age, without comorbidities (especially constipation, diabetes, Parkinson disease and spine injury) or medications (especially TCA and possibly CCB) and those with higher education level may represent an “easy-to prepare” group. It is interesting to investigate whether these patients may achieve adequate bowel preparation even without RE. Unfortunately, among the 18 studies focusing on investigating the effects of education on BP quality, none reported the results of bowel preparation in low-risk or high-risk patients. 

 We agree that it is a limitation since it is currently not possible to evaluate the effects of RE on BP quality in the population with low risks. The following contents were added into the Limitations (page 14 line 9-17): “Fourthly, patients with younger age, without comorbidities (especially constipation, diabetes, Parkinson disease and spine injury) or medications (especially tricyclic antidepressant (TCA) and possibly calcium channel blockers (CCB) and those with higher education level represent an relatively “easy-to-prepare” group (Am J Gastroenterol 2018;113:601-610). It is interesting to investigate whether these patients may achieve adequate bowel preparation even without RE. Unfortunately, among the 18 studies focusing on investigating the effects of RE on BP quality, none reported the results of bowel preparation in low-risk patients. Further studies are needed to investigate the effects RE on BP quality in the “easy to-prepare” population.”

3. The quality of the standard education varies in the different studies. It is possible that if the quality of the SE is low, such as written instructions only, the room for improvement may be greater than if the quality of the SE is better. Is it possible to group the studies in relation with the quality of the SE?

If it is not possible I will appreciate a comment in the limitations section.

Response: Thank you for the comments. We agree that the methods of SE may influence the quality of bowel preparation and the effects of RE. According to the suggestion, subgroup analysis were conducted and added into Results as follows (page 10, line 1-7): “SE methods. The methods of SE were described in detail in 16 studies, including 6 with written instructions alone, 1 with oral instructions alone and 9 using written plus oral instructions (Table 1). RE significantly improved the adequate rate in patients receiving only written instructions in the control group (1168/1308, 89.3% vs. 1005/1238, 81.2%; OR 2.29; 95%CI: 1.43-3.68; P<0.001). Similar results were found in patients receiving written plus oral instructions as the SE method (1300/1526, 85.2% vs. 963/1408, 68.4%; OR 2.77; 95%CI: 2.05-3.75; P<0.001).”

The methods of SE were illustrated in detail and supplemented in Table 1 as follows.

Table 1. Methods of RE and SE among different studies.

Study Method of RE Method of SE Adequate BP rate

Back, 2018 Audio-visual through smart phone Oral + written 96.4% vs. 73.6%

Calderwood, 2011 Visual aid Written 90.6% vs. 89.0%

Elvas, 2016 Additional personalized instruction Oral + written 62.1% vs. 35.4%

Ergen, 2016 Booklet NR 62.0% vs. 35.0%

Kang, 2015 Social media app Oral + written 90.0% vs. 75.6%

Lee, 2015 Telephone & SMS Oral + written 96.0% vs. 86.1%

Liu, 2018 Video plus retelling NR 90.0% vs. 75.1%

Liu, 2014 Telephone Oral + written 81.6% vs. 70.3%

Lorenzo, 2015 Smart phone app Written 100% vs. 96.1% 

Modi, 2009 Additional explanation Oral + written 69.0% vs. 57.5%

Park, 2015 SMS Written 79.4% vs. 57.8%

Park, 2016 Video Written 91.7% vs. 78.6%

Rice, 2016 Online video Oral + written 73.8% vs. 68.0%

Sharara, 2017 Smart phone app Written 82.5% vs. 77.5% 

Spiegel, 2011 New designed booklet Oral + written 75.8% vs. 46.3%

Tae, 2012 Cartoon visual aids Oral + written 93.1% vs. 81.6%

Walter, 2019 SMS Oral 90.7% vs. 80.6%

Wang, 2019 WeChat & SMS Written 87.5% vs. 65.4%

Abbreviations: RE, reinforced education; SE, standard education; BP, bowel preparation; SMS, short message service; NR, not reported

Minor

1. Originality. As the authors have stated there have been 4 previous meta-analysis exploring the association between an enhanced instruction for bowel prep and the bowel cleansing adequacy. Nevertheless the present study is the most updated analysis and it includes several RCTs in the last 4 years.

Response: Thank you for the comments. As the reviewer mentioned, compared with previous similar meta-analysis studies, the current study included another 4 latest RCTs and more patients (6536 vs. 2660-3795). The differences were described in detail in Introduction as follows (page 4, line 22-page5, line 2): “Four previous meta-analysis have been published to systemically compare the adequacy of BP among patients receiving enhanced instructions and standard education. The design and search strategies of these studies were different. Chang et al’s study was the first meta-analysis determining the effect of educational intervention on BP quality. Although 9 RCTs (n=2885) were included, 3 of them were abstracts. Desai et al enrolled 6 studies, which only detected smartphone applications on BP quality compared with standard education (n=810). Kurlander et al enrolled 7 studies with full articles (n=2660), however, 2 of them were not RCTs. In 2017, our team made comparisons of BP quality between patients receiving enhanced instructions plus regular instructions and regular instructions alone. 8 RCTs (n=3795) with full texts were enrolled. However, the literature search time were up to 2015. In the past 4 years, additional 10 high-quality clinical trials have been published. The results, however, seemed to be conflicting.”

2. There is a discrepancy between the 2 abstracts, in the first, line 8, it is stated that the literature search was done through March 2019. In the manuscript abstract, page 3, line 9, it says that the search was conducted until April 2019. Please correct it.

Response: Thank you for pointing out the mistake. The correct time is April 2019. The corresponding mistake has been corrected.

4. Page 6, lines 1,2 “diet restriction” is duplicated.

Response: Thank you for the suggestion. The duplicated “diet restriction” (pag6, line 1-2) has been deleted.

5. The effect of a reinforced education may be greater when the interval between the SE and the bowel preparation or the colonoscopy is longer, especially when the RE is made with a communicable tool. Is it possible to group the studies or the individual data to assess that variable?

Response: Thank you for the comments. We agree that RE may be more effective in patients with longer interval time from appointment day to the day of colonoscopy. Ten studies reported the details of interval time. Among them, 7 studies had shorter mean interval time (≤2 weeks) and 3 had longer (>2 weeks). Subgroup analysis showed that RE improved bowel preparation quality in both patients with longer (90.4% vs. 74.4%) or shorter (86.9% vs. 70.5%) mean interval time. The effects of RE seemed comparable between the two groups of patients (OR (95%CI): 2.79 (2.24-3.47) vs. 3.26 (1.31-8.14)).

Table. BP quality in patients with different interval time from appointment to colonoscopy.

Interval time from appointment to colonoscopy Study Adequate rate of bowel preparation OR (95%CI) P value

 RE SE 

≤2 weeks 7 86.9% (1365/1570) 70.5% (929/1317) 2.79 (2.24-3.47) <0.001

>2 weeks 3 90.4% (483/534) 74.4% (406/546) 3.26 (1.31-8.14) 0.010

Table. Details of interval time from appointment to colonoscopy among studies.

Study Interval time from appointment to colonoscopy

 Mean (days) RE SE

Back, 2018 17.3 NR NR

Calderwood, 2011 NR NR NR

Elvas, 2016 NR NR NR

Ergen, 2016 NR NR NR

Kang, 2015 14.6 14.6 14.5

Lee, 2015 5.4 5.4 5.4

Liu, 2018 3.3 3.4±0.2 3.3±0.8

Liu, 2014 3.4 3.4±0.8 3.5±0.9

Lorenzo, 2015 NR NR NR

Modi, 2009 NR NR NR

Park, 2015 6.02 5.02±2.02 7.03±1.38

Park, 2016 NR NR NR

Rice, 2016 30 NR NR

Sharara, 2017 NR NR NR

Spiegel, 2011 7 NR NR

Tae, 2012 NR NR NR

Walter, 2019 4 4 4

Wang, 2019 2 2 2

Reviewer #2: 

MAJOR

1) The authors need to improve the language in the manuscript. There are spelling mistakes (pay attention to the capital letters in the names, et al needs a dot after the word, like that “et al.”), and some difficult to understand sentences. For example:

i. Page 6, line 2 withdraw. You should change it for withdrawal.

ii. Page 6, line 6. You should change it for rate. I also suggest to add in this sentence, “The pooled rate of adequate bowel cleansing was …”

Response: Thank you for the suggestion. The errors mentioned by the reviewer had been corrected. In addition, a native English editor had been invited to review and modify the manuscript thoroughly. All grammatical and typing errors had been carefully checked and corrected. 

2) The figures and tables legend is not complete. Moreover, the references of the figures and supplementary figures don´t match in the text.

Response: We are sorry for the mistakes. The figures and tables legends were supplemented. Two authors had carefully and comprehensively double-checked the reference numbers in the manuscript, all figures and tables.

3) The authors have missed some articles in this meta-analysis, which fulfill the inclusion criteria. Do you have any reason to exclude them?

a. Galvez M, Zarate AM, Espino H, Higuera-de la Tijera F, Awad RA, Camacho S. A short telephone-call reminder improves bowel preparation, quality indicators and patient satisfaction with first colonoscopy. Endosc Int Open. 2017;5. E1172-E8.

b. Elvas L, Brito D, Areia M, Carvalho R, Alves S, Saraiva S, et al. Impact of Personalised Patient Education on Bowel Preparation for Colonoscopy: Prospective Randomised Controlled Trial. GE Port J Gastroenterol. 2017;24:22---30.

c. Prakash SR, Verma S, McGowan J, Smith BE, Shroff A, Gibson GH, et al. Improving the quality of colonoscopy bowel preparation using an educational video. Can J Gastroenterol. 2013;27:696---700.

Response: Thank you for telling us the important information. With careful evaluation, we agree that these three papers were relevant to reinforced education of bowel preparation. However, the primary outcome of this study was the rate of adequate bowel preparation. Only BBPS or OBPS score was reported for the evaluation of bowel preparation in the studies of Galvez et al. (Endosc Int Open. 2017;5. E1172-E8) and Prakash et al. (Can J Gastroenterol. 2013;27:696-700). From the original data of BBPS or OBPS score, the rates of adequate bowel preparation in could not be calculated. After full discussion, we decided not to include these two studies (Endosc Int Open. 2017;5. E1172-E8; Can J Gastroenterol. 2013;27:696-700) in this meta-analysis. 

Then with further enrollment of Elvas’ study (GE Port J Gastroenterol. 2017;24:22-30), totally 18 studies with 6536 patients were included for this meta-analysis. Data from all of the 18 studies were checked and re-analyzed carefully. All figures and tables were refreshed accordingly.

MINOR

4) It could help to the reader to explain the definition of standard instructions in the study selection.

Response: Thank you for the advice. Standard instructions mean oral instructions, written instructions or oral plus written instructions provided by physicians or nurses before colonoscopy. The definition was supplemented in Methods of the manuscript as follows (page 5, line 25-30): “SE meant oral instructions, written instructions or oral plus written instructions associated with bowel preparation, which was provided by physicians or nurses before colonoscopy. The contents of SE included diet restriction, the time and methods of drinking purgatives. RE referred to additional, enhanced instructions based on SE, which was realized by providing some certain of methods or tools. The contents of SE and RE were generally the same.”

5) Pag 7, paragraph 1. It could be better to add the reference after each bowel preparation scale to let the reader know which are the studies are you referring to.

Response: Thank you for this suggestion. Corresponding references have been added into Methods (page 6, line 1-4) as suggested. Here are the 5 original references related to BP scales.

1) BBPS: Lai EJ, Calderwood AH, Doros G, et al. The Boston bowel preparation scale: a valid and reliable instrument for colonoscopy-oriented research. Gastrointest Endosc 2009;69:620-5

2) OBPS: Rostom A, Jolicoeur E. Validation of a new scale for the assessment of bowel preparation quality. Gastrointest Endosc 2004;59:482-6.

3) UPAS: Modi C, Depasquale JR, Digiacomo WS, et al. Impact of patient education on quality of bowel preparation in outpatient colonoscopies. Qual Prim Care. 2009;17(6):397–404. 

4) HCS: Halphen M, Heresbach D, Gruss HJ, et al. Validation of the Harefield Cleansing Scale: a tool for the evaluation of bowel cleansing quality in both research and clinical practice. Gastrointest Endosc 2013;78:121-31.

5) Aronchick scale: Aronchick CA, Lipshutz WH, Wright SH, et al. A novel tableted purgative for colonoscopic preparation: efficacy and safety comparisons with Colyte and Fleet Phospho-Soda. Gastrointest Endosc 2000;52:346-52.

6) Pag 7, communicable and non-communicable tools. I suggest to add the definition and the reference to the correct supplementary table.

Response: Thank you for the suggestion. Communicable tools refer to the RE methods which allow patients to communicate with physicians or nurses if they had any questions about bowel preparation during the procedure of instruction or met problems during the preparation. Communicable tools in this study included social media application (WeChat) (Kang, 2015; Wang, 2019) and telephone call (Liu, 2014; Lee, 2015). According to the suggestion of the reviewer, the definition and corresponding references were added in Table 1 (page 16, line 2-18).

The following sentence was added into Results (page 8, line 11-20): “4 studies used communicable tools, including social media application (WeChat) (Kang, 2016; Wang, 2019) and telephone call (Liu, 2014; Lee, 2015). Through the communicable tools, patients could communicate with physicians or nurses if they had any questions about bowel preparation during the procedure of instruction or met problems during the preparation. By using communicable tools for RE, patients achieved better BP quality (1035/1166, 88.8% vs. 678/914, 74.2%; OR 2.84; 95%CI:1.97-4.11; P<0.001). In other 14 studies with non-communication tools as the RE methods, patients also showed a higher rate of adequate BP compared with the control group (1902/2198, 86.5% vs. 1681/2254, 74.6%; OR 2.52; 95%CI:1.92-3.30; P<0.001).”

7) Pag 8- Did you analyze the BP quality by using other purgatives as low volume preparations? 

Response: Thank you for the question. Several types of low-volume preparation are becoming popular in clinical practice due to its similar efficacy and lower adverse events compared with 4L PEG (Clin Gastroenterol Hepatol. 2019;S1542-3565(19)31246-7). It is interesting to investigate whether RE is useful for the improvement of BP quality in patients undergoing low-volume preparation.

Low-volume preparations were reported in 6 enrolled studies. The laxatives for preparation included 2L PEG+Asc (n=3), 2L PEG (n=3) and NaP or magnesium citrate (n=2). Subgroup analysis was conducted to evaluate the influences of RE on BP quality in the population undergoing low-volume preparations. 

The data was added into Results as follows (page 9, line 10-17): “Low-volume laxatives. Several types of low-volume preparations have recently been shown with similar efficacy and lower adverse events compared with 4L PEG (Endoscopy 2019; 51: 775–94; Clin Gastroenterol Hepatol. 2019;S1542-3565(19)31246-7). Here 6 studies with 2492 patients used low-volume preparations, including 2L PEG+Asc (n=3), 2L PEG or NaP or magnesium citrate (n=3). Compared with SE, RE showed higher rate of adequate BP in patients undergoing 2L PEG+Asc (576/609, 94.6% vs. 463/536, 86.4%; OR 2.84; 95%: 1.83-4.40; P<0.001) or 2L PEG (564/676, 83.4% vs. 451/671, 67.2%; OR 2.63; 95%CI: 1.75-3.97; P<0.001).”

8) It could be better to add in the results section some data regarding the diminutive adenoma detection rate as you discussed it latter.

Response: Thank you for the suggestion. The data of diminutive (1-5mm) adenoma detection rate were described in only one study. As the reviewer suggested, the following contents were added into Results (page 10, line 18-20): “Diminutive adenoma detection rate was reported in 1 study (Kang, 2016), which was also higher in the RE group (51/387, 13.2% vs. 30/383, 7.8%, P=0.019).”

Reviewer #3: 

Thank you for the opportunity to review this paper. This is an interesting manuscript presenting a results of a meta-analysis of randomized controlled trials on reinforced education improves the quality of bowel preparation for colonoscopy.

My review mainly concerns only the statistical aspects of the study. Some questions reported below were raised and in my view, it is not acceptable in this version for the publication in this journal.

Comments

1. Statistical analysis paragraph is a duplicate of the paragraph of the precedent paper [Guo X, Yang Z, Zhao L, et al. Enhanced instructions improve the quality of bowel preparation for colonoscopy: a meta-analysis of randomized controlled trials. Gastrointest Endosc 2017;85:90-97 e6.]. More detail are necessary. The meta-analysis was conduct on OR and on mean differences in the statistical analysis paragraph should be detailed.

Response: Thank you for the important comments. We totally agree with the reviewer that statistical methods should be described in more detail. 

According to the suggestion of the reviewer, the methods of Statistical analysis were modified as follows (page 6, line 24-page7, line 6): “All statistical analyses were performed using Review Manager (Revman, version 5.2) and Stata (version 12.0). If data from both intention-to-treat and per-protocol analyses were presented, the former were extracted and analyzed. Dichotomous data, including the rates of adequate bowel preparation, ADR or PDR, adverse events and diet compliance etc., were reported as odds ratio (OR) with 95% confidence interval (CI). Continuous data, including BBPS, OBPS, insertion and withdrawal time, were reported as standard mean difference (SMD) with 95%CI. Pooled estimates of OR or SMD were calculated using a random-effects model (M-H, heterogeneity), in which both within-study and between-study variations were considered (Control Clin Trials 1986;7:177-88.). Subgroup analysis were conducted according to the types of RE (communicable or not), evaluation tool (BBPS or OPBS), indication (screening or mixed) and preparation method (4L PEG, split-dose or low-volume laxatives). Statistical heterogeneity was accessed using the Cochrane Q test and by calculating the I2 value, with substantial heterogeneity defined as I2 greater than 50%, as described previously. A P value less than 0.05 was considered significant.”

2. Why in Figure 2 the OR estimated for Lee,2015 is different to the estimate of the previous paper? (3.91 vs 4.38)

Response: Thank you for the question. In the previous meta-analysis (Gastrointest Endosc 2017;85:90-97), the data of intervention group and control group were respectively 243/251 and 118/135 (OR, 4.38; 95%CI, 1.84-10.43). The data were retrieved from PP (per-protocol) analysis. In this study, the data were 243/253 and 118/137 (OR, 3.91; 95%CI, 1.76-8.68), which were retrieved from ITT (intention-to-treat) analysis. We believe ITT analysis might be better for the evaluation of the primary outcome. The results of ITT analysis were presented for all included studies in this study, which conferred on the differences of the results. 

The following sentence was added into Methods (page 6, line 26-27): “If data from both intention-to-treat and per-protocol analyses were presented, the former were extracted and analyzed.”

3. There are many subgroup analysis. The capabilites of subgroup analyses to detect meaningful differences between studies is often limited. Subgroup analyses also need sufficient power.

Response: Thanks for the comments. We admitted that it is one of limitations of this study. The following sentences were added into Limitations (page 14, line 19-24): “Last but not the least, to evaluate the effects of RE in different conditions, several subgroup analyses were performed. Although significant differences were found in most of the analyses (P<0.001). The power of the subgroup analyses may be not sufficient. The capabilities of subgroup analyses to detect meaningful differences between studies is often limited, thus it should be caution to explain the results.” 

Minor comments

1, There are many error of typing. Please, verify the manuscript for English and typos;

Response: Thank you for the suggestion. The manuscript had been thoroughly reviewed and modified by a native English editor. The grammatical errors and typing errors had been carefully checked and corrected.

2, p-value should be reported with three decimal number and with exact value; uniform the notation or P or p.

Response: Thank you for the suggestion. All P value were modified as with three decimal number or with exact value. “P”s were used as the uniform spelling.

3, In the funnel plot the line of the 95% CI can make easier to read the figure

Response: Thank you for the advice. The figure of funnel plot with both OR and 95%CI lines was re-produced by using Stata software. We agree that new figure was easier to read.

Additionally, the following sentence was added into Results (page 11, line 18-19): “Publication Bias. Begg’s test was conducted by Stata and the funnel plot showed no significant publication bias was found (P=0.950) (Supplementary figure 3).”

---

## [Decision Letter · Decision Letter 1]

13 Feb 2020

PONE-D-19-27355R1

Reinforced education improves the quality of bowel preparation for colonoscopy: an updated meta-analysis of randomized controlled trials

PLOS ONE

Dear Dr. Pan,

Thank you for submitting your manuscript to PLOS ONE. After careful consideration, we feel that it has merit but does not fully meet PLOS ONE’s publication criteria as it currently stands. Therefore, we invite you to submit a revised version of the manuscript that addresses the points raised during the review process.

ACADEMIC EDITOR: 

The authors properly answered most of the reviewers´ questions, however, further comments should be adressed . Please pay attention carefully to the statistical methods

We would appreciate receiving your revised manuscript by March 15th. To enhance the reproducibility of your results, we recommend that if applicable you deposit your laboratory protocols in protocols.io, where a protocol can be assigned its own identifier (DOI) such that it can be cited independently in the future. For instructions see: http://journals.plos.org/plosone/s/submission-guidelines#loc-laboratory-protocols

We look forward to receiving your revised manuscript.

Kind regards,

Antonio Z Gimeno-Garcia

Academic Editor

PLOS ONE

**Comments to the Author**

1. If the authors have adequately addressed your comments raised in a previous round of review and you feel that this manuscript is now acceptable for publication, you may indicate that here to bypass the “Comments to the Author” section, enter your conflict of interest statement in the “Confidential to Editor” section, and submit your "Accept" recommendation.

Reviewer #4: (No Response)

2. Is the manuscript technically sound, and do the data support the conclusions?

Reviewer #4: Partly

3. Has the statistical analysis been performed appropriately and rigorously? 

Reviewer #4: Yes

4. Have the authors made all data underlying the findings in their manuscript fully available?

Reviewer #4: Yes

5. Is the manuscript presented in an intelligible fashion and written in standard English?

Reviewer #4: No

6. Review Comments to the Author

Reviewer #4: This is a statistical review and I will focus on methods and reporting, mainly. Apologies, but this is the first time i see this paper.

Major

1) although for this journal novelty is irrelevant, the authors state that this is an updated meta-analysis in the abstract but they don't tell us why this is needed, what were the findings in the last one, and how many new studies have been added.

2) The language is quite poor, difficult to understand and not very academic. Consider the following passage: "For the primary endpoint, sensitivity analysis showed that a significantly better heterogeneity was noted (Heterogeneity: I2 from 56% to 34%) with removal of Calderwood’s study12. The following two reasons were considered: (1) it had the biggest sample size but resulted in a negative conclusion; (2) factors that caused the negative conclusion due to not providing patients with specific information of enhanced education, such as the solution of purgative use or diet restrictions." There is no better heterogeneity. the authors mean lower. also it is unclear what a negative conclusion is.

3) No information on publication bias tests and assessment in the methods section, only mentioned in the results section. Publication bias tests and plots only relevant if you have >10 studies otherwise underpowered to detect much and tend to lead to conclusions that are not justified http://www.ncbi.nlm.nih.gov/pubmed/11106885. Here, with your number of studies, you can state as a considerable strength.

4) Report the confidence intervals for I^2 (calculated using heterogi or metaan in Stata) as argued in http://www.ncbi.nlm.nih.gov/pubmed/17974687. A simple formula exists in the seminal 2002 Higgins paper that proposed I^2.

Minor

1) Some careful proof-reading is needed. Eg. the same sentence appears twice in the start of the methods section.

2) authors mean standardised rather than standard mean difference. Also are the relevant outcomes reported in different scales necessitating the use of SMDs? if you can avoid SMDs it is preferable since they are difficult to interpret, compared to MD, and a back-transformation may be needed to convery the effect to a scale the readers are familiar with.

3) Spell out Mantel-Haenszel.

4) Year may be worth considering in bias assessment, especially if you don't have enough studies for a formal test: http://www.ncbi.nlm.nih.gov/pubmed/25988604. With newer studies we would be more confident.

5) How was the random-effect model implemented, i.e. how was heterogeneity estimated? There are numerous ways to do so. Did they use the standard DerSimonian-Laird method? If so, please state so. Also there are better performing methods, for example please see https://www.ncbi.nlm.nih.gov/pubmed/28815652 (or http://www.ncbi.nlm.nih.gov/pubmed/23922860) and the metaan command in Stata where these are implemented (https://www.stata-journal.com/article.html?article=st0201).

6) Note that MH is traditionally a fixed effect approach and the random effects version in RevMan is an inverse variance-MH hybrid method, which has not been properly evaluated and is not avaialble elsewhere (except for metaan in Stata).

7) Cochran Q (i.e. chi-square) is notoriously underpowered to detect heterogeneity, especially for small meta-analyses http://www.ncbi.nlm.nih.gov/pubmed/9595615. I would not use

8) The authors do well to use RE models, since they outperform FE models in the presence of ANY heterogeneity. However, how did they assess heterogeneity? There are numerous ways to do so. Did they use the standard DerSimonian-Laird method? If so, please state so. Also there are better performing methods, for example please see http://www.ncbi.nlm.nih.gov/pubmed/23922860 and the metaan command in Stata where these are implemented.

7. PLOS authors have the option to publish the peer review history of their article (what does this mean?). If published, this will include your full peer review and any attached files.

Reviewer #4: No

---

## [Author Response · Author response to Decision Letter 1]

29 Mar 2020

Dear editor and reviewer,

We thank the editor and the reviewer for reviewing our Manuscript ID PONE-D-19-27355R1 entitled “Reinforced education improves the quality of bowel preparation for colonoscopy: an updated meta-analysis of randomized controlled trials”. We have revised the manuscript according to the constructive suggestions. A point-by-point response is included below. 

In response to the reviewers' comments, additional information has been added to the revised manuscript. These are marked by yellow highlights in the version with tracked changes. The version with all changes accepted (clean version) are also provided. Deleted text no longer appears in the final clean version.

Major

1) Although for this journal novelty is irrelevant, the authors state that this is an updated meta-analysis in the abstract but they don't tell us why this is needed, what were the findings in the last one, and how many new studies have been added.

Response: Thank you for the questions and suggestions. 

 There are some reasons why we carried out this updated meta-analysis. Firstly, there existed a number of new qualified studies related to reinforced education (ER) for bowel preparation (BP), which should be reviewed and analyzed. Secondly, with the development of science and techniques, tools and methods REs in recent years varied from the past, especially for the wide use of smartphones and their applications. Thirdly, some secondary outcomes enrolled a limited number of qualified studies and patients’ number, which may influence the accuracy and reliability of outcomes. Therefore, we decided to write an updated systemic review and meta-analysis.

 Compared to the previous meta-analysis conducted by our team (Gastrointest Endosc 2017;85:90-97 e6), this updated-meta analysis conducted the same conclusion for the primary outcome: BP quality. However, there were some new findings for secondary outcomes: 1) reinforced education (RE) improved both adenoma detection rate (ADR) (OR 1.35; 95%CI: 1.06-1.72; P=0.020) and polyp detection rate (PDR) (OR 1.24, 95%CI: 1.02-1.50; P=0.030), while it was negative in PDR (OR 1.25; 95% CI, 0.93-1.68; P=0.140) and not reported in ADR in the previous study; 2) patients in the RE group had a shorter insertion time (MD -0.76; 95%CI: -1.48-(-0.04); P=0.040), while there was no statistical difference in the previous study (MD, -0.57; 95% CI, -1.38 to 0.24; P=0.170); 3) less nausea/vomiting (OR 0.78; 95%CI: 0.64-0.97; P=0.020) and abdominal distension (OR 0.72; 95%CI: 0.68-0.92; P=0.020) were achieved in the RE group, while there was no statistical difference in the previous study for nausea/vomiting (OR 0.77; 95% CI, 0.60-0.99; P=0.050) and abdominal distention (OR 0.86; 95% CI, 0.66-1.12; P=0.260). The following sentences were added in Discussion section (line 3, page 12):“Although the primary outcome was similar to the four previous systemic review and meta-analyses, this updated meta-analysis conducted some new conclusions in secondary outcomes: 1) RE improved both ADR and PDR, which firstly demonstrated that patients receiving RE had a higher PDR in the form of systemic review and meta-analysis; 2) patients in the RE group had a shorter insertion time; 3) less nausea/vomiting and abdominal distension were achieved in the RE group.”

 In the Introduction section, we have described that:“In 2017, our team made comparisons of BP quality between patients receiving enhanced instructions plus regular instructions and regular instructions alone. 8 RCTs (n=3795) with full texts were enrolled. However, the literature search time were up to 2015. In the past 4 years, additional 10 high-quality clinical trials have been published14, 16, 19-21, 24, 29-32.”

2) The language is quite poor, difficult to understand and not very academic. Consider the following passage: "For the primary endpoint, sensitivity analysis showed that a significantly better heterogeneity was noted (Heterogeneity: I2 from 56% to 34%) with removal of Calderwood’s study12. The following two reasons were considered: (1) it had the biggest sample size but resulted in a negative conclusion; (2) factors that caused the negative conclusion due to not providing patients with specific information of enhanced education, such as the solution of purgative use or diet restrictions." There is no better heterogeneity. the authors mean lower. also it is unclear what a negative conclusion is.

Response: Thank you for the question and advice. Firstly, we apologize that the data of 56% and 34% were not correct, because they were the first edition’s data which included 17 studies and we forgot to correct. As we know, I2 from 25% to 50% was mild heterogeneity, I2 from 50% to 75% was moderate heterogeneity and >75% belongs to sever heterogeneity. In the first edition, only after extracting Calderwood’s study, I2 was 34%, while after extracting other studies one by one, all I2 values were >50%. Therefore, we detected the reasons of the decreasing of I2 from Calderwood’s study statistically and clinically. Among the all 18 included studies, 4 studies (Calderwood’s, Modi’s, Rice’s and Sharara’s study) conducted a negative conclusion that RE did not improve the quality of BP, while other 14 studies proved that RE could improve BP quality. And among all 18 included studies, Calderwood’s study had the biggest sample size (969/6536, 14.8%). So, it is worth detecting why this study drew a negative conclusion with the biggest sample size clinically. We found that the content of RE doctors provided to patients were not specific and quantified enough, e.g. the time when purgative was taken, the forbidden food and specific recommended food when undergoing BP.

 In fact, the correct overall I2 of 18 included studies was 53%. Sensitivity analysis was conducted by extracting studies one by one. After extracting Calderwood’s study and Back’s study, the I2 changed to 30% and 44% separately, while after the extraction of other studies one by one, all I2 were >50%. The following sentences were added in Method (line 9, page 11): “Sensitivity analysis For the primary endpoint, the I2 value of heterogeneity was 53%. Sensitivity analysis was conducted with the extraction of study one by one. It showed that after extracting Calderwood’s study and Back’s study, the I2 changed to 30% and 44% separately, while after the extraction of other studies one by one, all I2 values were >50%.”

3) No information on publication bias tests and assessment in the methods section, only mentioned in the results section. Publication bias tests and plots only relevant if you have >10 studies otherwise underpowered to detect much and tend to lead to conclusions that are not justified http://www.ncbi.nlm.nih.gov/pubmed/11106885. Here, with your number of studies, you can state as a considerable strength.

Response: Thank you for the suggestion. We have added publication bias tests and assessment in the Method section. The following sentence was added in Method section (line 4, page 7):“Publication bias was assessed by visual inspection of a funnel plot using Review Manager and was detected by Stata software”. The following sentence was added in Result section (line 9, page 11):“The funnel plots performed by Revman that was asymmetric (Supplementary figure 3). Begg’s test was conducted by Stata and the funnel plot showed no significant publication bias was found (P=0.950) (Supplementary figure 4).”

4) Report the confidence intervals for I^2 (calculated using heterogi or metaan in Stata) as argued in http://www.ncbi.nlm.nih.gov/pubmed/17974687. A simple formula exists in the seminal 2002 Higgins paper that proposed I^2.

Response: Thank you for the suggestion. We have carefully read the two passages you mentioned above, which made us had a deeper understanding of heterogeneity and its related values. Higgins’s study (Statist Med 2002;21:1539-1558) proposed three kinds of values to evaluate heterogeneity: H, R, and I^2. Although methods and formulas were introduced in the appendix of this study, we extraordinarily apologize that it was too difficult to calculate the CI of I^2. It seemed there was lack of formulas of 95%CI of I^2, which we finally failed to figure out. As stated in Ioannidis’s study (BMJ 2007;335(7626):914-6):“All statistical tests for heterogeneity are weak, including I^2. The clinical implications of this are considerable and must be examined on a case by case basis. Putting too much trust in homogeneity of effects may give a false sense of reassurance that one size fits all.

Minor

1) Some careful proof-reading is needed. Eg. the same sentence appears twice in the start of the methods section.

Response: Thank you for the suggestion and pointing out the mistake. The duplicated sentence about statistical analysis in the Methods section has been deleted. In addition, a native English editor had been invited to review and modify the manuscript thoroughly. All grammatical and typing errors had been carefully checked and corrected.

2) authors mean standardised rather than standard mean difference. Also are the relevant outcomes reported in different scales necessitating the use of SMDs? if you can avoid SMDs it is preferable since they are difficult to interpret, compared to MD, and a back-transformation may be needed to convery the effect to a scale the readers are familiar with.

Response: Thank you for the suggestion. We agree that using MD is more appropriate in this study. All continuous data have been changed to be reported as mean difference (MD) with 95%CI and the outcomes did not change. The sentence in Abstract section (line 14, page 3) have been changed as follows:” Continuous data were reported as mean difference (MD) with 95%CI”.

3) Spell out Mantel-Haenszel.

Response: Thank you for the suggestion. Because all outcomes were used RE models, which could use M-H method. The “M-H” in Method section has been deleted (line 31, Page 6).

4) Year may be worth considering in bias assessment, especially if you don't have enough studies for a formal test: http://www.ncbi.nlm.nih.gov/pubmed/25988604. With newer studies we would be more confident.

Response: Thank you for the suggestion. Among the included 18 RCTs, the earliest study was published in 2009 and most studies were reported in the last ten years. Publication years were summarized in Table 1.

Table 1. Time distribution of included studies

Year Studies

2001-2005 0

2006-2010 1

2011-2015 8

2016-2019 9

5) How was the random-effect model implemented, i.e. how was heterogeneity estimated? There are numerous ways to do so. Did they use the standard DerSimonian-Laird method? If so, please state so. Also there are better performing methods, for example please see https://www.ncbi.nlm.nih.gov/pubmed/28815652 (or http://www.ncbi.nlm.nih.gov/pubmed/23922860) and the metaan command in Stata where these are implemented (https://www.stata-journal.com/article.html?article=st0201).

Response: Thank you for the suggestion. In this meta-analysis, all primary and secondary outcomes were analyzed by Revman software, which could not use DerSimonian-Laird (D-L) method that is mostly used in random-effect models. As a supplementary part, the primary outcome was analyzed by Stata software using D-L method. The outcome of OR and 95%CI were the same between the two software (2.59 (2.09-3.19)). The forest plot of the primary outcome using Stata have been added as Supplementary Figure 5.

6) Note that MH is traditionally a fixed effect approach and the random effects version in RevMan is an inverse variance-MH hybrid method, which has not been properly evaluated and is not avaialble elsewhere (except for metaan in Stata).

Response: Thank you for the reminder and suggestion. As a supplementary part, the primary outcome was analyzed by Stata software using D-L method. The outcome of OR and 95%CI were the same between the two software (2.59 (2.09-3.19)).

7) Cochran Q (i.e. chi-square) is notoriously underpowered to detect heterogeneity, especially for small meta-analyses http://www.ncbi.nlm.nih.gov/pubmed/9595615. I would not use 

Response: Thank you for the advice. We agree that Cochran Q (i.e. chi-square) test is less statistically valid than I2. “using the Cochrane Q test” has been deleted in Abstract and Methods sections.

8) The authors do well to use RE models, since they outperform FE models in the presence of ANY heterogeneity. However, how did they assess heterogeneity? There are numerous ways to do so. Did they use the standard DerSimonian-Laird method? If so, please state so. Also there are better performing methods, for example please see http://www.ncbi.nlm.nih.gov/pubmed/23922860 and the metaan command in Stata where these are implemented.

Response: Thank you for the question and suggestion. In fact, in this meta-analysis, the I2 were both 53% using FE and RE models, belonging to moderate heterogeneity. Considering the method and tools of reinforced education varied from all included studies, which may lead to the heterogeneity, therefore, RE model was used. For heterogeneity test, firstly, subgroup analyses were made. Secondly, sensitivity analysis was conducted with the extraction of studies one by one, which finally did not show a significantly better heterogeneity. Thirdly, D-L method was used in Stata software, which conducted the same conclusion with using Revman software.

---

## [Editor Report · Decision Letter 2]

3 Apr 2020

Reinforced education improves the quality of bowel preparation for colonoscopy: an updated meta-analysis of randomized controlled trials

PONE-D-19-27355R2

Dear Dr. Pan,

We are pleased to inform you that your manuscript has been judged scientifically suitable for publication and will be formally accepted for publication once it complies with all outstanding technical requirements.

With kind regards,

Antonio Z Gimeno-Garcia

Academic Editor

PLOS ONE

---

## [Editor Report · Acceptance letter]

8 Apr 2020

PONE-D-19-27355R2 

Reinforced education improves the quality of bowel preparation for colonoscopy: an updated meta-analysis of randomized controlled trials 

Dear Dr. Pan:

I am pleased to inform you that your manuscript has been deemed suitable for publication in PLOS ONE. Congratulations! Your manuscript is now with our production department. 

With kind regards,

on behalf of

Dr. Antonio Z Gimeno-Garcia 

Academic Editor

PLOS ONE